# Deep Learning Automated Segmentation for Muscle and Adipose Tissue from Abdominal Computed Tomography in Polytrauma Patients

**DOI:** 10.3390/s21062083

**Published:** 2021-03-16

**Authors:** Leanne L. G. C. Ackermans, Leroy Volmer, Leonard Wee, Ralph Brecheisen, Patricia Sánchez-González, Alexander P. Seiffert, Enrique J. Gómez, Andre Dekker, Jan A. Ten Bosch, Steven M. W. Olde Damink, Taco J. Blokhuis

**Affiliations:** 1Department of Traumatology, Maastricht University Medical Centre+, 6229 HX Maastricht, The Netherlands; jan.ten.bosch@mumc.nl (J.A.T.B.); taco.blokhuis@mumc.nl (T.J.B.); 2Department of Surgery, NUTRIM School of Nutrition and Translational Research in Metabolism, Maastricht University Medical Centre+, 6229 HX Maastricht, The Netherlands; r.brecheisen@maastrichtuniversity.nl (R.B.); steven.oldedamink@maastrichtuniversity.nl (S.M.W.O.D.); 3Department of Radiation Oncology (MAASTRO), GROW School for Oncology and Development Biology, Maastricht University Medical Centre+, 6229 HX Maastricht, The Netherlands; leonard.wee@maastro.nl (L.W.); andre.dekker@maastro.nl (A.D.); 4Clinical Data Science, Faculty of Health Medicine and Lifesciences, Maastricht University, Paul Henri Spaaklaan 1, 6229 GT Maastricht, The Netherlands; 5Biomedical Engineering and Telemedicine Centre, ETSI Telecomunicación, Center for Biomedical Technology, Universidad Politécnica de Madrid, 28040 Madrid, Spain; p.sanchez@upm.es (P.S.-G.); ap.seiffert@upm.es (A.P.S.); enriquejavier.gomez@upm.es (E.J.G.); 6Centro de Investigación Biomédica en Red de Bioingeniería, Biomateriales y Nanomedicina (CIBER-BBN), 28029 Madrid, Spain; 7Department of General, Visceral and Transplantation Surgery, RWTH University Hospital Aachen, 52074 Aachen, Germany

**Keywords:** sarcopenia, deep learning neural network, automated segmentation, computed tomography

## Abstract

Manual segmentation of muscle and adipose compartments from computed tomography (CT) axial images is a potential bottleneck in early rapid detection and quantification of sarcopenia. A prototype deep learning neural network was trained on a multi-center collection of 3413 abdominal cancer surgery subjects to automatically segment truncal muscle, subcutaneous adipose tissue and visceral adipose tissue at the L3 lumbar vertebral level. Segmentations were externally tested on 233 polytrauma subjects. Although after severe trauma abdominal CT scans are quickly and robustly delivered, with often motion or scatter artefacts, incomplete vertebral bodies or arms that influence image quality, the concordance was generally very good for the body composition indices of Skeletal Muscle Radiation Attenuation (SMRA) (Concordance Correlation Coefficient (CCC) = 0.92), Visceral Adipose Tissue index (VATI) (CCC = 0.99) and Subcutaneous Adipose Tissue Index (SATI) (CCC = 0.99). In conclusion, this article showed an automated and accurate segmentation system to segment the cross-sectional muscle and adipose area L3 lumbar spine level on abdominal CT. Future perspectives will include fine-tuning the algorithm and minimizing the outliers.

## 1. Introduction

Body composition, defined as the amount of fat relative to the amount of muscle in the body, has been linked to clinical outcomes in a number of conditions [1,2,3]. Sarcopenia, as the progressive loss of skeletal muscle mass (SMM), is considered to be a muscle disease inducing muscle impairment [4]. Sarcopenia was initially described in the elderly population as loss of muscle mass and strength with advancing age [5], but is proven to be a problem in various patient groups. Poor muscle status and increased adipose tissue is associated with significant health effects and increased morbidity and mortality [1]. The clinical application of altered body composition and sarcopenia in risk assessment and treatment decision, therefore, warrants early identification.

CT imaging, on the level of the third lumbar vertebra, is the current gold standard for quantification of muscle mass [5,6,7,8,9,10]. Parameters are obtained based on the analysis of a single slice CT scan. Shen et al. demonstrated that skeletal muscle area (SMA) which is the cross-sectional muscle area calculated at the level of the third lumbar vertebra, is able to correctly estimate total body muscle mass [11]. To relate SMA to total muscle mass, it is normalized to the square of height, resulting in the skeletal muscle index (SMI) [12].

Muscle radiation attenuation, calculated as the average Hounsfield Unit [HU) of the cross-sectional muscle area, is a measurement of muscle density, with lower values indicating higher muscle fat content. Segmentation of muscle area in the determination of sarcopenia on CT is done manually, by tracing the muscle group margins on axial sections. Although the inter- and intra-observer agreement for this method is excellent [13], it is a labor-intensive process, which requires diagnostic accuracy and time-consuming involvement from a radiologist. The required time investment limits the use of CT for routine clinical measurement of muscle mass, and thereby, not widely used for early identification of patients at risk.

Deep learning techniques are emerging to support diagnoses with high accuracy, enhancing the speed of image interpretation and, thus, improve the clinical efficiency for a wide range of medical tasks. For example, recent studies showed improved and accurate body composition on CT [14,15]. For muscle mass measurement, recent studies created and validated automated segmentation of the abdominal muscle on manually extracted CT-image at the L3 level [16,17,18,19]. These studies, including patients with varying medical conditions, showed detailed information regarding severity of sarcopenia. However, patients had been excluded when a motion or scatter artefact was present, resulting in a highly restricted population; moreover, they were not multi-center studies and all studies suffered from small sample sizes [16,17].

Following severe trauma, abdominal CT scans are routinely obtained in the emergency evaluation upon admission. These CT scans provide accurate information regarding internal injuries, but need to be quickly and robustly delivered and must not cause a delay in definitive trauma care. This implies that trauma CT scans often contain motion or scatter artefacts, incomplete vertebral bodies or contain arms that influence image quality [20]. Another limiting factor is the heterogeneity in polytrauma patients, such as the different trauma mechanisms, medical conditions and ages. As recently shown, the prevalence of sarcopenia in older trauma patients is significant [21], thereby putting these patients at risk for complications.

Since early identification of sarcopenia in these patients would be clinically valuable for trauma case management, and since an abdominal CT scan is often present, the aim of this study was to investigate whether trauma-CT images can be used in a new deep learning algorithm for quantification of muscle mass. The necessary requirement was rapid automated segmentation and on-the-spot computation of body composition. In this article, we report on the initial development and external generalizability test of a deep learning neural network that was trained on thousands of CTs of abdominal cancer surgery patients. We hypothesized that by allowing the network to train on a large multi-institutional dataset, including a wide range of scanner manufacturers and image acquisition settings, the automated segmentation would be more widely generalizable. We then tested whether the deep learning model could segment and quantify truncal musculature at L3 lumbar vertebral level on abdominal CT scans of polytrauma patients, for the potential use in rapid clinical detection of sarcopenia.

## 2. Materials and Methods

### 2.1. Training and Validation Set: Cancer Surgery Cases

A deep learning neural network was trained on a multi-center collection of 3413 abdominal cancer surgery subjects to automatically segment truncal musculature, subcutaneous adipose and visceral adipose at the L3 lumbar vertebral level. Several different cohorts of patient groups were used to train and validate the algorithm. Two cohorts of patients with colorectal liver metastases were selected in the UK [22]. One cohort of patients with colorectal liver metastases was selected from Zuyderland Medical Center. Two cohorts with patients with ovarian cancer consists of data from five centers in the Netherlands [23,24]. One cohort consists of 304 patients treated for a resectable PDAC of the pancreatic head in the Uniklinik Aachen (Aachen, Germany) and MUMC+ (Maastricht, Netherlands).

### 2.2. Independent Test Set: Polytrauma Cases

Patients with an injury severity score (ISS) greater or equal to 16 who presented at a single level-1 trauma center between 2015 and 2019, and with an abdominal computed tomography (CT) axial image (supine position at third lumbar level) at admission were extracted from a regional trauma registry. Included patients were selected as a related dataset that would pose an independent and challenging validation for a deep-learning algorithm trained on the aforementioned surgical CTs.

As a reference for segmentation, author L.A. manually edited the margins of the L3 muscle (L3M), intramuscular adipose tissue (IMAT), Visceral Adipose Tissue (VAT) and Subcutaneous Adipose Tissue (SAT) generated by the TomoVision (Magog, QC, Canada) software “sliceOmatic” (version 5.0 Rev 6b) using semi-automated threshold-based segmentation on a window of −29 to +150 Hounsfield Units (HU) [25]. L3M included rectus abdominis, external and internal abdominal obliques, transversus abdominis, erector spinae, psoas major and minor and quadratus lumborum. All segmentations were cross-checked by authors J.T.B. and T.J.B., surgeons with 5 and 15 years of clinical experience, respectively, as being consistent with the Alberta protocol [25].

From the segmented areas, we computed Skeletal Muscle Radiation Attenuation (SMRA), Skeletal Muscle Index (SMI), VAT index (VATI) and SAT index (SATI) in a well-established manner [12,26]. There was no exclusion of CT images on basis of either motion artefacts, scatter artefacts, signal-to-noise ratio, abdominal wall hernia or presence of hands/arms in the field of view.

### 2.3. Deep Learning

A deep-learning neural network (DLNN) for multi-label segmentation of L3M, VAT and SAT was based on a standard two-dimensional U-Net [27], with only minor adjustments to fit this task. Pre-processing of CT images was done with a widely-used deep learning procedure by first clipping the CT image intensities to fall between −200 HU and +200 HU. A schematic block diagram of the DLNN architecture is given as Appendix A. Briefly, blocks in the downsizing path each consisted of two convolution layers with pixel padding of 1 to maintain consistent size of input to the next layer. Batch normalization [28] was performed after every convolution layer with max-pooling [29] in between every block. Ridge regression regularization [30] was applied to reduce the effect of over-fitting. The upsizing path comprised transposed convolutions concatenated with some downsizing path features introduced via skip-connections. All except the final convolution layer used parametric rectified linear units (PReLUs) [31] as activation function and ADAM optimizer [32]. The final convolution layer was followed by softmax activation to extract probability density maps for four segmentation labels (L3M, VAT, SAT or other). Hyper-parameter fine-tuning was performed exclusively on the training dataset through trial-and-error; the final hyper-parameters used for this work are stated in Appendix A.

### 2.4. Performance Statistics

For each case in the trauma dataset, the geometric similarity of deep learning automatic segmentation was validated against the reference source of truth using a Dice Similarity Coefficient (DSC). We computed four body composition metrics—SMRA, SMI, VATI and SATI—from the deep-learning generated segmentations to compare against the reference values. Agreement was quantified using Lin’s Concordance Correlation Coefficient (CCC) [33] and Bland-Altman’s Limits of Agreement interval (LOA) [34]. The Bland-Altman plots are presented in the Appendix A. Additionally, as a rough estimate of inter-observer divergence, we used SMI values from our previously published paper [21] and compared these to SMI from the abovementioned reference L3M segmentations. All statistical analysis was performed in R [35] (version 4.0.3) using the “epiR” library [36] (accessed on 15 January 2021).

## 3. Results

A wide range of abdominal surgery cases from 31 distinct cancer centers made up the training dataset. The general case mix summary has been given in Table 1. Scanners from at least four different manufacturers were included, but possibly more, since vendor information was not always retained in the DICOM metadata. Median values for imaging parameters were 120 kVp, 5 mm reconstructed slice thickness and 0.768 mm × 0.768 mm axial spacing and 90% of patients were scanned in the feet-first supine position. Arms and hands had been kept out of the L3 slice field of view, being held on or folded over the chest. Image quality metrics of Signal-Noise-Ratio (SNR) and Contrast-Noise-Ratio (CNR) are also reported in Table 1. The SNR was calculated for skeletal muscle compared to background, and the CNR was calculated for fat with respect to muscle.

In total, 233 polytrauma cases were suitable for inclusion as the test dataset. A flowchart of case attrition numbers has been included as Figure 1, and the general case characteristics were also shown in Table 1. The average age was 74 (range 10–88) years and the mean BMI was 29.5 (range 13.2–45.7) kg/m^2^. There were 156 male patients (67%) and 77 female patients. All cases were imaged according to the polytrauma protocol settings. The patients were presented head first supine position, slice thickness 1–3 mm, 120 kVp and 254 mAs/slice with an iodine-based contrast. Two scanners were used: Philips brilliance 64-slice scanner and Siemens SOMATOM Definition Flash CT Scanner.

### 3.1. Similarity of Segmentation

The overall geometric accuracy of automated segmentation compared to the reference segmentation was good; median DSC (and interquartile range) for L3M, VAT and SAT were 0.926 (0.866–0.959), 0.951 (0.888–0.974) and 0.953 (0.916–0.975), respectively. Perfect agreement implied a DSC of 1, whereas no overlap at all between the reference and automated segmentation implied a DSC of 0.

For added transparency of our results, the distribution of DSC in each of the three body components are shown in a box-and-whisker plot as Figure 2. It is clear that, while overall geometric performance is good, there are several cases with low DSC. There was only a single case where the automated segmentation failed to produce any output at all; hence, its DSC was exactly 0.

From all 233 individual results of the automated segmentation, we arbitrarily selected six as representative examples of the overall geometric findings. These are presented in Figure 3. The majority of cases with poor automated segmentation performance coincided with unusually noisy or poor-quality CT images. The second most common cause of discrepancies was the appearance of hands and/or arms in the CT field of view, though these were only rarely present in the training dataset. Much rarer trauma cases showing poor automated segmentation results involved the presence of external foreign objects adjacent to the abdomen, atypical anatomy (such as extremely low muscle mass) or clear signs of subcutaneous emphysema.

### 3.2. Agreement Analysis

Quantitative assessment of Lin’s CCC and Bland-Altman’s LOA intervals indicated generally good results. The concordance was generally very good for the body composition indices of SMRA (CCC = 0.92), VATI (CCC = 0.99) and SATI (CCC = 0.99), whereas perfect agreement implied a CCC of 1. Visual confirmation of these results is shown as the concordance plots in Figure 4. The overall result for SMI was degraded (CCC = 0.71) in comparison to the other indices. From Figure 4b, it is clear that hands/arms in the CT view leads to a consistent and systematic over-estimation of the SMI value in the polytrauma cases relative to the reference truth.

Exploring the differences between CT slices with or without hands/arms in the field of view, we see from Table 2 that the results of SMRA and SMI with hands/arms in the field of view tended to decrease the overall CCC. The SMI agreement for only those cases with hands/arms in the field of view was significantly reduced (subgroup CCC = 0.58). Similar trends were shown by the bias correction factor (which is the scale shift required to dispose the datapoints around the ideal concordance line) and LOA intervals; the bias corrections were close to 1 (which implied perfect agreement) and the LOA intervals of agreement were close to 0 (which implied perfect agreement), except for SMI and particularly for SMI in the subgroup with hands/arms included in the CT.

We additionally compared the reference truth in this work with previously reported SMI values [21]. This provided a very limited but nonetheless indicative estimate of potential disagreement between independent observers working on the same data at different times (see again Table 2). The interobserver CCC for SMI was 0.88, which was slightly better in contrast to the automated SMI CCC of 0.83 on just those images without hands/arms, but it was not statistically significantly better.

Sarcopenia was determined using cutoff values for SMI as described by Prado et al. (52.4 cm^2^/m^2^ and 38.5 cm^2^/m^2^ for males and females, respectively). The accuracy of sarcopenia classification based on the automated segmentation relative to reference segmentation was 77% (sensitivity 59%, specificity 96%). The numbers of automated false positives and false negatives were 5/233 (2%) and 48/233 (21%), respectively. Considering only CTs with no hands or arms in the field of view, the accuracy was 90%, sensitivity 86% and specificity 94%, with 4/131 (3%) false positives and 9/131 (7%) false negatives.

For comparison, a previous investigator who independently segmented the polytrauma dataset compared to the present reference segmentation had an accuracy of 82%, sensitivity of 66% and specificity of 98%, with false positives in 2/233 (1%) and false negatives in 40/233 (17%), though there was no difference in the human-to-human discriminative performance with or without hands/arms in the CT field of view.

## 4. Discussion

This work showed that an automated deep learning segmentation algorithm trained on a massive and diverse multi-center surgical dataset yielded overall good geometric agreement with respect to a human expert reference segmentation, for adipose and muscle tissues, on abdominal CT scans at the L3 level. Geometric performance was degraded due to the presence of hands in the polytrauma test dataset, which was only very rarely found in the surgical training dataset. Geometric agreement was a surrogate measure of clinical relevance; in tests for agreement against human reference-based SMRA, SMI, VATI and SATI values, the deep learning algorithm performed well overall on CCC and LOA measures. The algorithm consistently and systematically over-estimated the SMI, as was confirmed in the geometric comparisons.

This constitutes the first suggestion that our deep learning model could potentially be used in the future, subject to further development and extensive validation, to provide rapid body composition quantification, and for supporting the diagnosis of sarcopenia in trauma patients, from a generic L3 abdominal axial CT. To the best of our knowledge, no deep learning system has yet been robustly tested to this degree, using related but wholly independent CT images from a completely different clinical setting.

Other studies approached the automated segmentation of muscle, VAT and SAT within the L3 vertebra region in a variety of ways, also including deep learning. These reported DSC ranged from 0.85–0.99 [16,17,18,19,37]; however, good comparisons between those studies cannot be made due to the use of similar patient cohorts in development and accuracy testing and restrictive sample sizes. These models were able to extract semantic information, overall muscle shape and adipose tissue; nevertheless, these studies analyzed computed tomography scans, which were made in a controlled (non-trauma) setting [16,17,38].

Given the fact that after severe trauma, abdominal CT scans are almost routinely obtained in the emergency evaluation, and the features in CT scans to diagnose sarcopenia are presently not being used because of the time-consuming effort, the need for automated segmentation is evident. Although these CT scans were briskly and non-selectively obtained, with often some motion or scatter artefacts, imperfect vertebral bodies or arms/hands in the view, and contain artefacts and/or poorer image quality, the concordance was generally very good for the body composition indices of SMRA (CCC = 0.92), VATI (CCC = 0.99) and SATI (CCC = 0.99). The interobserver CCC in SMI was 0.88 (95% confidence interval 0.86–0.91). This provides a very limited but indicative estimate of potential disagreement between independent observers working on the same data at different times. The agreement in SMI due to the automated segmentation initially looks much poorer, but closer inspection of Table 2 shows that the CT scans containing hands or arms in the field of view drags the overall result downwards. In the sub-group without hands/arm visible, the deep learning performance was almost comparable with human-to-human performance. However, a complete study of inter- and intra-observer divergences was presently not feasible using the current datasets. Our results are interesting because the unedited outcomes of our algorithm (CCC = 0.83) approach but are still below the estimate of concordance between human experts working independently (CCC = 0.88), so we acknowledge that our model should be further improved by more training.

The accuracy of the algorithm was 77% overall, which is only marginally poorer than human-to-human accuracy (82%), but we expect this to rise if 90% if future development work can overcome the over-segmentation of hands and arms, where present.

Assuming a qualified physician remains in supervision of the procedure, such an algorithm could provide clinically relevant information supporting the rapid detection of sarcopenia in the trauma setting. Sarcopenia is associated with negative health outcomes and is, therefore, an important piece of differential prognostic information in clinical practice. For a clinically useful algorithm that identifies patients with a high risk of sarcopenia, we would want the number of false negatives to be as low as possible, since this represents a detrimental health condition going undetected during care. Conversely, we might be willing to tolerate slightly more false positives, since treatment of sarcopenia involves better nutrition and more functional activity [39], which is unlikely to harm anyone. Every case of muscle decline that is detected through such an algorithm might then reduce the severity of sarcopenia, and thereby, its associated negative health outcomes.

Our deep learning algorithm was trained using a diverse geographical cohort of patients that did not contain any polytrauma patients. While the algorithm suggests potential for rapid clinical assessment of body composition, we note some important limitations in the present work. We have noted that this algorithm systematically over-estimates muscle area, because it tends to include the muscle tissue in the hands and arms, if these are included in the field of view. Additionally, overlapping adjacent internal organs with muscle, and CT Hounsfield Units being similar between some organs and muscle, also lead to a degree of misidentification as muscle.

Due to time pressure for this trauma protocol CT setting, there were several imaging artefacts which affected the segmentation that were not corrected by re-scanning. We chose to include all of these challenging cases, because they are fully representative to real trauma cases. Some segmentations contained a CT “streak” scatter artefact near the spine that led to internal organs and adipose being mislabeled. Foreign objects lying in the field of view sometimes created strong scatter artefacts that led to misclassification of subcutaneous adipose tissue as muscle. Noisy or poor-quality CT scans resulted in dispersed spots of undetected adipose tissue and muscle. Rarely, post-traumatic subcutaneous emphysema at the L3 level also led to missed detection of subcutaneous adipose tissue.

After locating the L3 slice, segmenting the muscle tissue alone using the HU thresholding software required approximately 20 to 30 min per slice [38]. From the same starting point, the deep learning model segmented muscle, subcutaneous fat and visceral fat in an average of 0.4 s per slice in total (on our device; for hardware specifications, please see the hyperparameters section in the Appendix A). We acknowledge that segmentation timing may vary greatly on different hardware. The principal time efficiency gained by a deep learning approach is because the model executes with no additional human interaction (other than choosing the L3 slice), and our reported results do not contain any kind of post-segmentation editing by a human user. Most of the time consumed during the HU thresholding approach was due to the human operator adjusting the thresholds and growing the body compartments, then afterwards manually adjusting the results by hand. The latter is arguably the most accurate; however, it is comparatively time-consuming, though our results show that human-made segmentations are themselves susceptible to disagreement and uncertainty.

In regards to future work, we propose two promising areas for making significant improvements: firstly, we will improve the overall agreement by retraining the neural network to exclude the hands. Secondly, we plan to test a workflow that includes automatic detection of the L3 slice from whole body axial CT scans. Ideally, our system should be extended to incorporate whole body analysis rather than just one axial slice, providing rapid and accurate characterization of comprehensive body morphometric parameters. Further work may also be done to comprehensively search for globally optimal hyper-parameter settings, as these may also lead to some additional improvements in geometric accuracy.

## 5. Conclusions

This article showed that a deep learning U-Net-based algorithm was able to segment cross-sectional muscle and adipose area at L3 lumbar spine level on abdominal CT of realistic trauma patients. Clinically relevant body composition metrics were computed from the automated segmentation, which showed good agreement and clinically congruent decisions compared against a reference human expert segmentation. Further work on algorithm development should be able to improve the geometric accuracy, quantitative agreement and diagnostic discrimination of sarcopenia in our clinical setting, as well as produce whole-body CT scans for comprehensive morphological analysis.

## Figures and Tables

**Figure 1 sensors-21-02083-f001:**
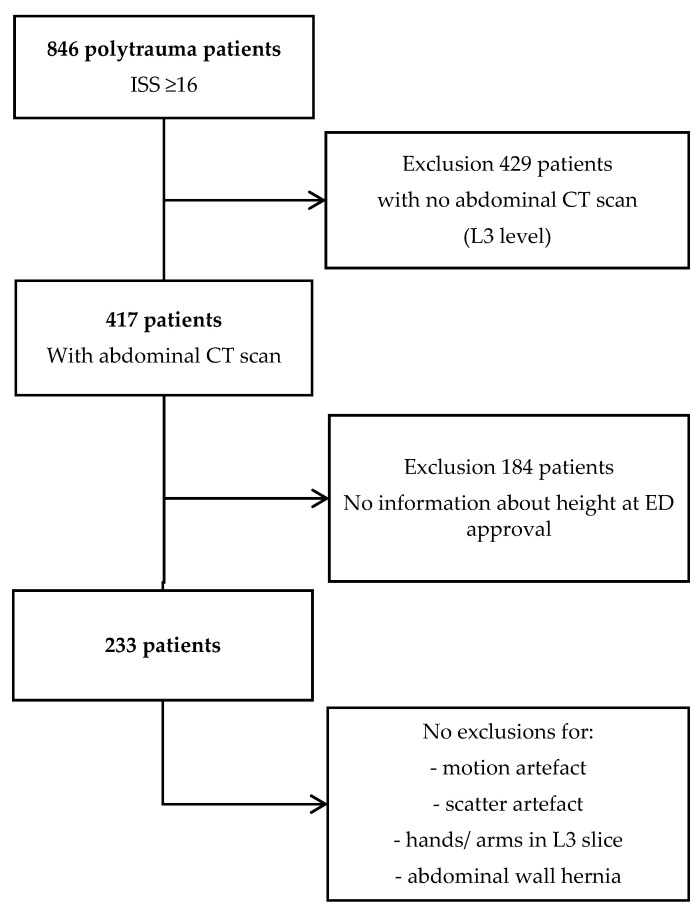
Flowchart of test case attrition numbers from a trauma case registry. ISS = Injury Severity Score.

**Figure 2 sensors-21-02083-f002:**
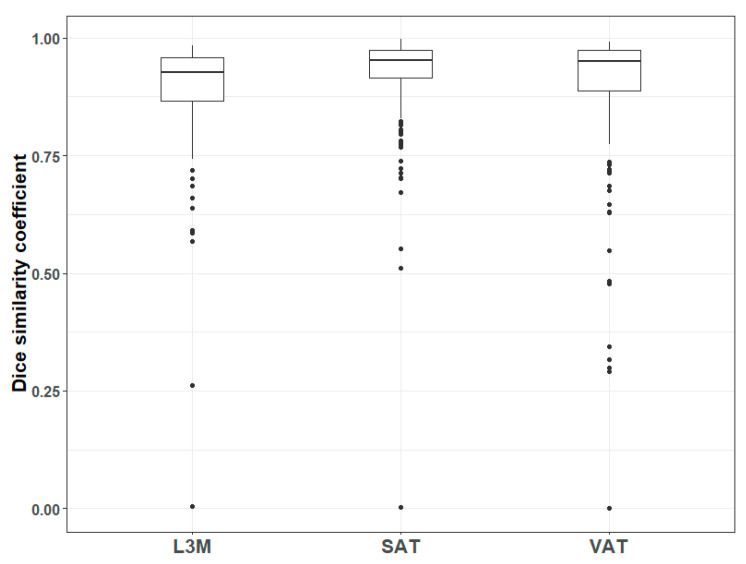
A box and whisker plot illustrating the distribution of Dice Similarity Coefficient (DSC) for the L3 lumbar muscle (L3M), subcutaneous adipose tissue (SAT) and visceral adipose tissue (VAT). The thicker horizontal bar represents the median value, the edges of the box represent the upper and lower 25 percentiles and the thin vertical lines represent the limits of the 1 percentiles. Data outliers beyond these limits have been plotted in the figure as small solid circles.

**Figure 3 sensors-21-02083-f003:**
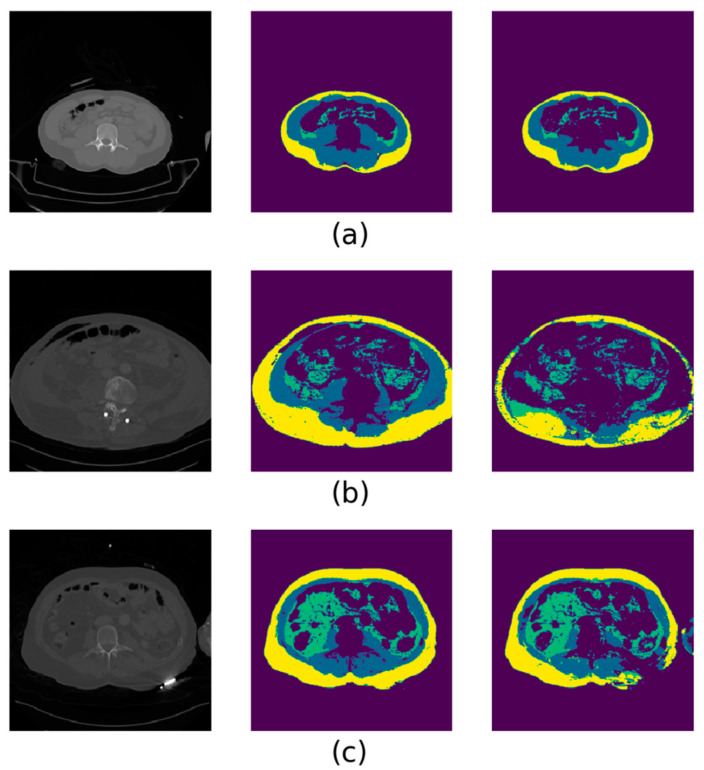
Selected examples of deep-learning automated segmentation results, with representative errors represented. In each image, the original L3 CT slice is shown on the left, the ground truth segmentation in the middle and the automated segmentation on the right. The color scheme is as follows—yellow: subcutaneous adipose tissue; blue: lumbar muscle; green: visceral adipose tissue. (**a**) An automatic segmentation that would be deemed clinically acceptable. (**b**) A CT “streak” scatter artefact near the spine that led to internal organs and adipose being mislabeled. (**c**) A case of an unknown foreign object lying under the left dorsolateral side of the patient, creating strong scatter artefacts that led to the misclassification of subcutaneous adipose as muscle. (**d**) A common event in the trauma dataset that was only rarely seen in the training dataset, i.e., hands and arms in the CT field of view being misclassified as lumbar muscle. (**e**) A noisier CT image than usual, resulting in spots of undetected adipose and muscle. (**f**) A rare case of post-traumatic subcutaneous emphysema, leading to missed detection of subcutaneous adipose.

**Figure 4 sensors-21-02083-f004:**
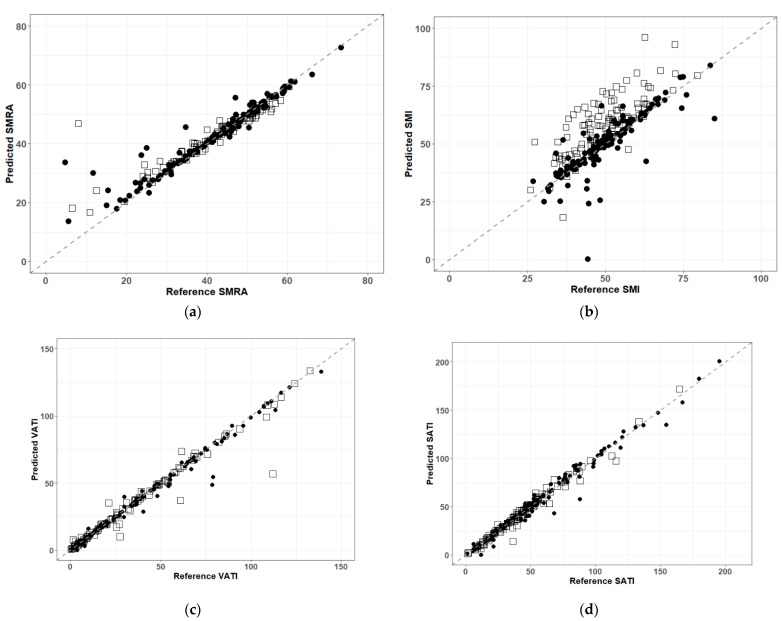
Concordance correlation plots for (**a**) SMRA, (**b**) SMI, (**c**) VATI and (**d**) SATI. Values from abdominal CT images that contained hands and/or arms in the field of view were plotted with an open square, whereas images without hands/arms were plotted with a solid circle. A dashed 45-degree line running through (0,0) is provided as a guide to the eye. Points lying further from the dashed line implied greater disagreement with respect to body indices calculated from the reference truth segmentations.

**Table 1 sensors-21-02083-t001:** Patient characteristics of the entire cohort (*n* = 3413), training (*n* = 2730, 80%), validation set (*n* = 683, 20%) and the test subset (n = 233).

Patient Characteristics	Training Set							Test Set
Total	*n* = 3413							*n* = 233
Mean SNR	0.9							1.1
Mean CNR	2.0							0.8
Disease	Colorectal cancer				Ovarian cancer		Pancreatic cancer	Polytrauma patients
Study	newEPOC *	FROG’s *	Zuyderland	Zuyderland	MUMC **		Aachen ***	MUMC
Year	2007–2012	2017–2019	2013–2017	2013–2017	2002–2015	2004–2014	2010–2017	2015–2019
Total	153	804	226	1587	216	123	304	233
Male	-	374 (58%)	-	883 (56%)	0	0	161 (53%)	156 (66.9%)
Female	-	430 (42%)	-	704 (44%)	216 (100%)	123 (100%)	143 (47%)	77 (33.1%)
Age (years)	-	25–95 (mean 68.2)	-	32–93 (median: 70)	30–101	39–86	mean 67.7 (SD 10.2)	10–88 (mean 74)
BMI (kg/m^2^)	-	13.7–58.1 (mean 26.4)	-	1553 (median: 26)	-	-	25.4 (SD 4.2)	13.2–45.7 (mean 29.5)

* Bristol, Poole, Bournemouth, Royal Marsden, Surrey, Portsmouth, Velindre, Sheffield, Imperial Charing X, Imperial St Mary, Christie, Southend, Yeovil, North Middlesex, Southampton, Guys, Aintree, Winchester, Cambridge, Princess Alexandra, Bedford, Salisbury, UCL, Basingstoke, Pennine; ** MUMC, Nijmegen, Bernhoven, St. Jansdal, Ede; *** Aachen and MUMC.—No exact values, patient demographics are in line with other data presented.

**Table 2 sensors-21-02083-t002:** Summary of the agreement statistics, quantified as the Concordance Correlation Coefficient (CCC), bias correction error and Bland-Altman Limits of Agreement (LOA) interval. The best unbiased estimates are given with the 95% confidence intervals given in parentheses. The units of LOA intervals are the same as the body composition index. CCC and bias correction are dimensionless.

	Concordance Correlation	Bias Correction Error	Limits of Agreement
**SMRA**			
All	0.92 (0.91–0.94)	0.98	−0.99 (−9.3–7.3) HU
Sub: hands	0.89 (0.85–0.92)	0.96	−1.0 (−10–8.2) HU
Sub: no hands	0.95 (0.93–0.96)	0.99	−0.97 (−8.5–6.6) HU
**SMI**			
All	0.71 (0.64–0.76)	0.93	−4.0 (−21–13) kg·m^−2^
All (interobs.)	0.88 (0.86–0.91)	0.97	−2.7 (−12–6.3) kg·m^−2^
Sub: hands	0.58 (0.48–0.67)	0.74	−9.4 (−25–6.2) kg·m^−2^
Sub: no hands	0.83 (0.77–0.88)	0.99	−0.69 (−28–29) kg·m^−2^
**VATI**			
All	0.99 (0.98–0.99)	1.00	0.98 (−9.7–12) kg·m^−2^
Sub: hands	0.98 (0.97–0.98)	1.00	0.87 (−12–14) kg·m^−2^
Sub: no hands	0.99 (0.99–0.99)	1.00	1.1 (−7.0–9.1) kg·m^−2^
**SATI**			
All	0.99 (0.98–0.99)	1.00	0.29 (−9.8–10) kg·m^−2^
Sub: hands	0.99 (0.98–0.99)	1.00	0.00 (−9.2–9.2) kg·m^−2^
Sub: no hands	0.99 (0.98–0.99)	1.00	0.52 (−10–11) kg·m^−2^

## Data Availability

No new data were created or analyzed in this study. Data sharing is not applicable to this article.

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
