# Peer review of "Deep Learning Automated Segmentation for Muscle and Adipose Tissue from Abdominal Computed Tomography in Polytrauma Patients"

_sensors, 2021, doi:10.3390/s21062083_

Round 1

Reviewer 1 Report

Muscle, subcutaneous adipose tissue, and visceral adipose tissue were segmented with UNET on the CT image at the level of L3. My review is based on clinical viewpoint and ethical viewpoint. Because of the latter, I recommend rejection.  

Major points

  • “Institutional Review Board Statement: The investigations were carried out following the rules of the Declaration of Helsinki of 1975, revised in 2013. Surgical cancer data was obtained with agreement from the study principal investigators. Ethical review and approval were waived for the polytrauma cases, since these anonymized data extracted with permission from a regional registry.“ I cannot accept this. If the datasets are not freely available through Internet, IRB approval should be obtained from each institution.

  • Maybe, UNET of this paper processed the CT image only at the level of L3. Therefore, clinicians must select the CT image at the level of L3, before using the UNET. If so, the UNET is not fully automatic. This is major limitation of this study.

  • “As reference for  segmentation,  author  LA  manually  edited  the  margins  of  the  L3 muscle (L3M), intramuscular adipose tissue (IMAT), Visceral Adipose Tissue (VAT) and Subcutaneous Adipose Tissue (SAT) generated by the TomoVision (Magog, Quebec, Canada) software “sliceOmatic” (version 5.0 Rev 6b) using semi-automated threshold-based  segmentation on a window of -29 to +150 Hounsfield Units (HU) (25). “ If the UNET is not fully automatic, I speculate that sliceOmatic is enough for evaluating sarcopenia clinically.

  • The UNET segmented all muscles in the CT images. I am not sure that this segmentation results are useful for evaluation of sarcopenia clinically. This should be investigated based on previous studies. Ref. 21 is not enough.

Minor points

  • DSC and DCS were used in this paper. Please use DSC.

  • If CCC value is less than 0.9, the agreement is considered as poor (see the following papers). Therefore, for SMI, both interobserver value and UNET’s value are considered as poor agreement. This should be described.

   https://pubmed.ncbi.nlm.nih.gov/22618522/

   https://pubmed.ncbi.nlm.nih.gov/27058307/

  • Partly, color of bones is black in Figure 2. Why? Generally, color of bones is white in CT.

  • Please add Bland-Altman plots.

Author Response

Dear reviewer,

Thank you for providing feedback on our manuscript. We appreciate your comments and believe that they have helped us improve the manuscript. Those changes are marked with “track changes” within the manuscript. Please see below, in italics, for a point-by-point response to the comments. All page numbers and line numbers hereafter refer to the revised manuscript file.

 R1Q1) “Institutional Review Board Statement: The investigations were carried out following the rules of the Declaration of Helsinki of 1975, revised in 2013. Surgical cancer data was obtained with agreement from the study principal investigators. Ethical review and approval were waived for the polytrauma cases, since these anonymized data extracted with permission from a regional registry. “I cannot accept this. If the datasets are not freely available through Internet, IRB approval should be obtained from each institution.

R1A1) Thank you for this comment and the opportunity to clarify. For cancer surgery studies, the original research data were collected with individual IRB approval for each study; a list of institutional review board (IRB) reference numbers have been provided to the Editor in a separate letter. The principal investigators of the original studies provided the anonymized CT images and segmentation masks, from which re-identification was not feasible. For extraction and re-analysis of polytrauma case data, approval was given by the Maastricht University Medical Centre IRB (reference METC 2018-0756). Requirement for informed consent was waived because of the retrospective nature of this study. This is in accordance with the Instructions to Authors for this journal.

We have clarified the IRB statement in the revised manuscript:

[Section: Informed consent, Page 13, Line]

Institutional Review Board Statement: The investigations were carried out following the rules of the Declaration of Helsinki of 1975, revised in 2013. For cancer surgery studies, the original research data were collected with individual IRB approval for each study; a list of institutional review board (IRB) reference numbers have been provided to the Editor. The principal investigators of the original studies provided the anonymized CT images and segmentation masks. No new data were created or analyzed by the authors of this study; data sharing is not applicable to this article. For extraction and re-analysis of polytrauma case data, approval was given by the Maastricht University Medical Centre IRB (reference METC 2018-0756). Requirement for informed consent was waived because of the retrospective nature of this study.”

R1Q2) Maybe, UNET of this paper processed the CT image only at the level of L3. Therefore, clinicians must select the CT image at the level of L3, before using the UNET. If so, the UNET is not fully automatic. This is major limitation of this study.

R1A2) Thank you for this comment. We did not mean for this to be the understanding of our current work. In the text, we tried to inform that the only step we are automating is the segmentation. For any of the segmentation methods discussed in this manuscript, the L3 slices have to be selected first.

However, we wish to point out that selection of images is a distinct and separate step from the segmentation process. The specific scope of this paper is the demarcation of subcutaneous fat, visceral fat and skeletal muscle from a selected L3 slice, which we have achieved using no additional human intervention. Furthermore, the results we have reported are the direct results of the algorithm – there has been no post-processing editing by any human user.

As mentioned in the discussion {page 15, line 390-410}, present work is under way towards automated selection of the L3 slice, as well as further re-training of the model to exclude hands from the muscle segmentation. In the longer term, our system will be extended to incorporate whole body analysis rather than just a single axial slice, providing rapid and accurate characterization of comprehensive body morphometric parameters. This is however out of scope of the present manuscript. The prospects for our future research does not prevent us from reporting the present work, because it is a complete and stand-alone scientific investigation of high quality in its own right.

To comply with the wish of Reviewer 2, we have added the following to the Discussion, that will also serve here to emphasize our intended scope and improve clarity:

{Discussion, page 15, line 390-410},

“After locating the L3 slice, segmenting the muscle tissue alone using the HU thresholding software required approximately 20 to 30 minutes per slice (38). From the same starting point, the deep learning model segmented muscle, subcutaneous fat and visceral fat in an average of 0.4 seconds per slice in total (on our device; for hardware specifications please see the hyperparameters section in the Supplementary Materials). We acknowledge that segmentation timing may vary greatly on different hardware. The principal time efficiency gained by a deep learning approach is because the model executes with no additional human interaction (other than choosing the L3 slice) and our reported results do not con-tain any kind of post-segmentation editing by a human user. Most of the time consumed during the HU thresholding approach was due to the human operator adjusting the thresholds and growing the body compartments, then afterwards manually adjusting the results by hand. The latter is arguably the most accurate however comparatively time-consuming, though our results show that human-made segmentations are themselves susceptible to disagreement and uncertainty.

In regards to future work, we propose two promising areas for making significant improvements; first to improve the overall agreement by re-training the neural network to exclude the hands. Secondly, we plan to test a workflow that includes automatic detection of the L3 slice from whole body axial CT scans. Ideally, our system should be extended to incorporate whole body analysis rather than just one axial slice, providing rapid and accurate characterization of comprehensive body morphometric parameters.

In interest of avoidance of potential confusion about the scope of work, we also made the following minor changes to improve clarity:

{Page 3, line 94-96}

“In this article, we report on initial development and external generalizability test of a deep learning neural network that was trained on thousands of CTs of abdominal cancer surgery patients.”

{Page 15, line 370-372}

“Every case of muscle decline that is detected through such an algorithm might then reduce the severity of sarcopenia and thereby its associated negative health outcomes.”

R1Q3) “As reference for segmentation, author LA manually edited  the  margins  of  the  L3 muscle (L3M), intramuscular adipose tissue (IMAT), Visceral Adipose Tissue (VAT) and Subcutaneous Adipose Tissue (SAT) generated by the TomoVision (Magog, Quebec, Canada) software “sliceOmatic” (version 5.0 Rev 6b) using semi-automated threshold-based  segmentation on a window of -29 to +150 Hounsfield Units (HU) (25). “If the UNET is not fully automatic, I speculate that sliceOmatic is enough for evaluating sarcopenia clinically.

R1A3) This has been discussed at length in R1A2 above, and also in compliance with Reviewer 2 (R2A3). This has been addressed in the same way as for the aforementioned comments.

{Discussion, page 15, line 390-410},

“After locating the L3 slice, segmenting the muscle tissue alone using the HU thresholding software required approximately 20 to 30 minutes per slice (38). From the same starting point, the deep learning model segmented muscle, subcutaneous fat and visceral fat in an average of 0.4 seconds per slice in total (on our device; for hardware specifications please see the hyperparameters section in the Supplementary Materials). We acknowledge that segmentation timing may vary greatly on different hardware. The principal time efficiency gained by a deep learning approach is because the model executes with no additional human interaction (other than choosing the L3 slice) and our reported results do not con-tain any kind of post-segmentation editing by a human user. Most of the time consumed during the HU thresholding approach was due to the human operator adjusting the thresholds and growing the body compartments, then afterwards manually adjusting the results by hand. The latter is arguably the most accurate however comparatively time-consuming, though our results show that human-made segmentations are themselves susceptible to disagreement and uncertainty.

In regards to future work, we propose two promising areas for making significant improvements; first to improve the overall agreement by re-training the neural network to exclude the hands. Secondly, we plan to test a workflow that includes automatic detection of the L3 slice from whole body axial CT scans. Ideally, our system should be extended to incorporate whole body analysis rather than just one axial slice, providing rapid and accurate characterization of comprehensive body morphometric parameters.

R1Q4) The UNET segmented all muscles in the CT images. I am not sure that this segmentation results are useful for evaluation of sarcopenia clinically. This should be investigated based on previous studies. Ref. 21 is not enough.

R1A4) Thank you for sharing your opinion. The procedure used in this study adheres to a well-established procedure for the evaluation of sarcopenia clinically, that has indeed been widely published and widely reported. Studies do in fact provide evidence that the measurement derived from an abdominal axcial CT image does estimate total body skeletal muscle and adipose tissues for a wide range of healthy adults and patients. These studies provide the justification, clinical outcome and importance of diagnosing sarcopenia from CT imaging. The following references have all been used in the revised manuscript. [Lee, 2019 #44][Sergi, 2016 #45][Engelke, 2018 #46][Sharma, 2015 #47] [Prado, 2008 #48] [Baracos, 2018 #49][Shen, 2004 #52]

R1Q5) DSC and DCS were used in this paper. Please use DSC.

R1A5) Thank you for pointing this out this typo. This has been done in the revised manuscript on the following pages and lines.

[Section Similarity of segmentation, Line 203, Page 6]

[Section Similarity of segmentation, Line 205, Page 6]

[Section Similarity of segmentation, Line 206, Page 6]

[Section Similarity of segmentation, Line 207, Page 6]

[Section Similarity of segmentation, Line 209, Page 6]

[Section Similarity of segmentation, Line 211, Page 6]

[Section Figure 2, Similarity of segmentation, Line 223, Page 7]

R1Q6) If CCC value is less than 0.9, the agreement is considered as poor (see the following papers). Therefore, for SMI, both interobserver value and UNET’s value are considered as poor agreement. This should be described.

  • https://pubmed.ncbi.nlm.nih.gov/22618522/
  • https://pubmed.ncbi.nlm.nih.gov/27058307/

R1A6) Thank you for sharing these papers. Though they are interesting and well written in their own right, we respectfully argue that they not appropriate to this particular situation.

These papers are about iterative mathematical image reconstruction from multiple projections, where a single source of truth exists. This is not the case for medical image segmentation problems because there is not one single undisputed source of truth; there are in fact multiple sources of truth that do not completely agree with one another. Recommendations in the existing literature, for example Schober et al [Schober, 2018 #136] states clearly: “Cutoff points (of what is meant by a “strong” or “weak” association) are arbitrary and inconsistent and should be used judiciously …. Rather than using oversimplified rules, we suggest that a specific coefficient should be interpreted as a measure of the strength of the relationship in the context of the posed scientific question.”

In the context of our scientific question, it is not the point to show that deep learning models would have a concordance of 1.00, because even human experts do not perfectly agree with each other. We conclusively showed that the concordance of our results for SMI, SMRA, VATI and SATI compared well with the interobserver concordance, especially when we also provided the reader with all the transparent information about the subgroups (hands versus no hands in the field of view). Here, the important context that our manuscript highlights is that, given its known limitations, our model is already showing concordance that approaches the same strength of relationship as exists between two independent human experts, as required by Schober et al. We want our readers to be able to appreciate this, so wewould like to point out that describing all SMI results, both interobserver and model-derived, as “poor” does not lead the reader to a complete understanding of the context of AI in this clinical topic.

R1Q7) Partly, color of bones is black in Figure 2. Why? Generally, color of bones is white in CT.

R1A7) Dear reviewer, thank you for this comment. This is the same as question R2Q1 from the previous reviewer. We acknowledge that we did not explain this properly in the deep learning description in the Methods section. The well-established procedure in deep learning is to restrict the Hounsfield Units to a fixed window, in this case -200 HU to +200 HU. The bone signal lies outside this range. The CT image we provided in Figure 2 (revised Figure 3) was the CT image after clipping to [-200 HU, 200 HU]. To avoid potential for confusion, we have put the original unclipped CT images into revised Figure 3

We have also added the following text to the Methods section, to address our inadequate description:

[Methods, Line 138/139, Page 4]

“Pre-processing of CT images kept to widely-used deep learning procedure by first clipping the CT image intensities to fall between -200 HU and +200 HU.”

R1Q8) Please add Bland-Altman plots.

R1A8) Dear reviewer, this has been done in the Supplemental Materials as Figure S3.

Reviewer 2 Report

Thank you for giving me the opportunity to review your manuscript entitled ‘Deep learning automated segmentation for muscle and adipose tissue from abdominal computed tomography in polytrauma patients’ submitted to the journal of Sensors.

The authors evaluated and concluded that they could provide an automated and accurate segmentation system to segment the cross-sectional muscle and adipose area L3 lumbar spine level on abdominal CT.

There are some comments on this manuscript. I hope these comments can improve their manuscript.

Major

Materials and methods section

  1. Looking at the CT in figure 2 (left column), the bone signal has been removed, but I can't find a description of it. Please mention it.

  1. The target image quality (i.e. SNR, CNR, or SD and so on) in each research should be briefly described in this manuscript and Table. I also recommend the author should mention the target image quality of their trauma CTs. Image quality was not able to estimate by tube voltage and current of CT scan parameters.

Results section

  1. The authors described that manual segmentation was time-consuming. How long did their segmentation system to segment the muscle and report it? Please mention it.

Minor

  1. There are two figures 2 in the manuscript, which are duplicated. Please correct it.

  1. Sample images are difficult to see because they are too small. I would like you to make them bigger.

  1. In concordance correlation plots, open square and solid circle are small. I would like you to make them bigger.

Author Response

Dear reviewer,

Thank you for providing feedback on our manuscript. We appreciate your comments and believe that they have helped us improve the manuscript. Those changes are marked with “track changes” within the manuscript. Please see below, in italics, for a point-by-point response to the comments. All page numbers and line numbers hereafter refer to the revised manuscript file.

R2Q1) Looking at the CT in figure 2 (left column), the bone signal has been removed, but I can't find a description of it. Please mention it.

R2A1) Dear reviewer, thank you for this comment. This is the same as question R1Q7 from the previous reviewer. We acknowledge that we did not explain this properly in the deep learning description in the Methods section. The well-established procedure in deep learning is to restrict the Hounsfield Units to a fixed window, in this case -200 HU to +200 HU. The bone signal lies outside this range. The CT image we provided in Figure 2 (revised Figure 3) was the CT image after clipping to [-200 HU, 200 HU]. To avoid potential for confusion, we have put the original unclipped CT images into revised Figure 3

We have also added the following text to the Methods section, to address our inadequate description:

[Methods, Line 138/139, Page 4]

 “Pre-processing of CT images kept to widely-used deep learning procedure by first clipping the CT image intensities to fall between -200 HU and +200 HU.”

R2Q2) The target image quality (i.e. SNR, CNR, or SD and so on) in each research should be briefly described in this manuscript and Table. I also recommend the author should mention the target image quality of their trauma CTs. Image quality was not able to estimate by tube voltage and current of CT scan parameters.

R2A2) Dear reviewer, thank you for this comment. We have now added two additional rows into Table 1 under the section heading of image quality metrics. In the Results section, we have also added the following line of text:

[Results, Line 177-179, Page 5]

“Image quality metrics of Signal-Noise-Ratio (SNR) and Contrast-Noise-Ratio (CNR) are also reported in Table 1. The SNR was calculated for skeletal muscle compared to background, and the CNR was calculated for fat with respect to muscle.”

R2Q3) The authors described that manual segmentation was time-consuming. How long did their segmentation system to segment the muscle and report it? Please mention it.

R2A3) Thank you for raising this important question. We have now added the following discussion about timing and potential efficiency (with caveats about timing on other hardware systems) in the Discussion section:

{Discussion, page 15, line 390-410},

“After locating the L3 slice, segmenting the muscle tissue alone using the HU thresholding software required approximately 20 to 30 minutes per slice (38). From the same starting point, the deep learning model segmented muscle, subcutaneous fat and visceral fat in an average of 0.4 seconds per slice in total (on our device; for hardware specifications please see the hyperparameters section in the Supplementary Materials). We acknowledge that segmentation timing may vary greatly on different hardware. The principal time efficiency gained by a deep learning approach is because the model executes with no additional human interaction (other than choosing the L3 slice) and our reported results do not con-tain any kind of post-segmentation editing by a human user. Most of the time consumed during the HU thresholding approach was due to the human operator adjusting the thresholds and growing the body compartments, then afterwards manually adjusting the results by hand. The latter is arguably the most accurate however comparatively time-consuming, though our results show that human-made segmentations are themselves susceptible to disagreement and uncertainty.

In regards to future work, we propose two promising areas for making significant improvements; first to improve the overall agreement by re-training the neural network to exclude the hands. Secondly, we plan to test a workflow that includes automatic detection of the L3 slice from whole body axial CT scans. Ideally, our system should be extended to incorporate whole body analysis rather than just one axial slice, providing rapid and accurate characterization of comprehensive body morphometric parameters.

R2Q4) There are two figures 2 in the manuscript, which are duplicated. Please correct it.

R2A4) Dear reviewer, thank for you finding this typo. The figure numbers have been corrected in the revised manuscript.

R2Q5) Sample images are difficult to see because they are too small. I would like you to make them bigger.

R2A5) Dear reviewer, we agree. We have tried to enlarge the figures and reduce the white space around the images. This has been done in the revised manuscript as Figure 3.

R2Q6) In concordance correlation plots, open square and solid circle are small. I would like you to make them bigger.

R2A6) Dear reviewer, we have enlarged the symbols for the data points as requested. This has been done in the revised manuscript as the updated Figure 4.

Round 2

Reviewer 1 Report

1

For example, Figure 6 of “7. Engelke K, Museyko O, Wang L, Laredo J-D. Quantitative analysis of skeletal muscle by computed tomography imaging—State of the art. Journal of orthopaedic translation. 2018;15:91-103.” shows “The left CT image at the level of L3; the right segmentation: subcutaneous adipose tissue (blue), paraspinal muscle(brown), abdominal muscle (orange), intramuscular adipose tissue (green), visceral fat (yellow), and visceral organs (grey). “

Therefore, abdominal muscle and paraspinal muscle are separately segmented in the Figure 6 of ref. 7. Why do not authors sperate these muscle in the authors’ UNET for evaluating sarcopenia clinically?

2

R1A6) Thank you for sharing these papers. Though they are interesting and well written in their own right, we respectfully argue that they not appropriate to this particular situation.” I cannot accept this reply. The criteria of CCC value are widely accepted. For example, the following paper used the same criteria (Concordance of the Recently Published Body Adiposity Index With Measured Body Fat Percent in European-American Adults, https://www.ncbi.nlm.nih.gov/pmc/articles/PMC3988697/) . Please rewrite the Result section based on the criteria. See the following papers.

https://pubmed.ncbi.nlm.nih.gov/22618522/

https://pubmed.ncbi.nlm.nih.gov/27058307/

https://pubmed.ncbi.nlm.nih.gov/24838736/

https://www.ncbi.nlm.nih.gov/pmc/articles/PMC3988697

Author Response

Dear reviewer,

Thank you for providing second feedback on our manuscript. We appreciate your new comments and believe that they have helped us improve the manuscript. Those changes are marked with “track changes” within the manuscript. Please see below, in italics, for a point-by-point response to the comments. All page numbers and line numbers hereafter refer to the revised manuscript file.

R1Q1) For example, Figure 6 of “7. Engelke K, Museyko O, Wang L, Laredo J-D. Quantitative analysis of skeletal muscle by computed tomography imaging—State of the art. Journal of orthopaedic translation. 2018; 15:91-103.” shows “The left CT image at the level of L3; the right segmentation: subcutaneous adipose tissue (blue), paraspinal muscle(brown), abdominal muscle (orange), intramuscular adipose tissue (green), visceral fat (yellow), and visceral organs (grey). “

Therefore, abdominal muscle and paraspinal muscle are separately segmented in the Figure 6 of ref. 7. Why do not authors sperate these muscle in the authors’ UNET for evaluating sarcopenia clinically?

R1A1) Thank you for this comment and the opportunity to clarify this issue. In our work, we followed the widely used Alberta protocol for Tomovision SliceOmatic (1), in which the total muscle area in the third lumbar level is segmented. In our understanding, it  can be considered the current gold standard for quantification of muscle mass, as shown in the provided references (2-8). Although Engelke et al. segmented the paraspinal and abdominal muscle separately, we have seen that the most common used preference is the segmentation of total muscle area on the third lumbar level. Engelke et al. have not stated that this separation is better than total segmentation of muscle area. Furthermore, we believe that separating abdominal and paraspinal muscle cannot be meaningful, since it would have no further clinical impact when calculating the Skeletal Muscle Index (9).

R2Q2) “R1A6) Thank you for sharing these papers. Though they are interesting and well written in their own right, we respectfully argue that they not appropriate to this particular situation.” I cannot accept this reply. The criteria of CCC value are widely accepted. For example, the following paper used the same criteria (Concordance of the Recently Published Body Adiposity Index With Measured Body Fat Percent in European-American Adults, https://www.ncbi.nlm.nih.gov/pmc/articles/PMC3988697/) . Please rewrite the Result section based on the criteria. See the following papers.

https://pubmed.ncbi.nlm.nih.gov/22618522/

https://pubmed.ncbi.nlm.nih.gov/27058307/

https://pubmed.ncbi.nlm.nih.gov/24838736/

https://www.ncbi.nlm.nih.gov/pmc/articles/PMC3988697

R2A2) Thank you for sharing these papers and the opportunity to improve the manuscript. These all use the CCC criteria and are a measurement whereas perfect agreement would be signified by all data lying on the 1:1 line.

In the first revision, we tried to explain  the reasons why we think a more open approach is more appropriate and cited a strong defense of the idea to not to apply arbitrary cut-offs for poor or good(10). McBride from 2005 is a proposal paper for a government report that, to the best of our knowledge, has not been tested by scientific review. We think that the other papers cited do not give explanations of why McBride 2005 cut-offs are appropriate or not. This forces words in our mouths that we do not deserve, because it implies that we are saying that standard practice in the huge amount of body composition literature using CT analysis is therefore “poor”, which will then be rejected by the other experts in our field. So as to not create prejudice, we think it is better to simply report the numbers and make clear the context, and let the reader judge for themselves.

We believe that our results would be of interest because the unedited outcomes of our algorithm (CCC=0.83) approaches but are still below the estimate of concordance between human experts working independently (CCC = 0.88). However, we acknowledge that our model should be further improved by more training, which would be performed in future works.

We have discussed your response with the co-author group and understand your opinion. Although it may differ from ours, we understand that we should adjust the discussion to improve clarity: Thus, we have included the following text:

{Page 15, line 358} “Our results are interesting because the unedited outcomes of our algorithm (CCC=0.83) approaches but is still below an estimate of concordance between human experts working independently (CCC = 0.88), so we acknowledge that our model should be further improved by more training.”

  1. sliceOmatic. Alberta Protocol. 11 February  2017 [20-01-2021]. https://tomovision.com/Sarcopenia_Help/index.htm].
  2. Chung H, Cobzas D, Birdsell L, Lieffers J, Baracos V. Automated segmentation of muscle and adipose tissue on CT images for human body composition analysis: SPIE; 2009.
  3. Lee K, Shin Y, Huh J, Sung YS, Lee I-S, Yoon K-H, et al. Recent Issues on Body Composition Imaging for Sarcopenia Evaluation. Korean journal of radiology. 2019;20(2):205-17.
  4. Sergi G, Trevisan C, Veronese N, Lucato P, Manzato E. Imaging of sarcopenia. European journal of radiology. 2016;85(8):1519-24.
  5. Sharma P, Zargar-Shoshtari K, Caracciolo JT, Fishman M, Poch MA, Pow-Sang J, et al., editors. Sarcopenia as a predictor of overall survival after cytoreductive nephrectomy for metastatic renal cell carcinoma. Urologic Oncology: Seminars and Original Investigations; 2015: Elsevier.
  6. Baracos VE, Mazurak VC, Bhullar AS. Cancer cachexia is defined by an ongoing loss of skeletal muscle mass. Annals of palliative medicine. 2018;8(1):3-12.
  7. Shen W, Punyanitya M, Wang Z, Gallagher D, St.-Onge M-P, Albu J, et al. Total body skeletal muscle and adipose tissue volumes: estimation from a single abdominal cross-sectional image. Journal of applied physiology. 2004;97(6):2333-8.
  8. Sinelnikov A, Qu C, Fetzer DT, Pelletier JS, Dunn MA, Tsung A, et al. Measurement of skeletal muscle area: Comparison of CT and MR imaging. Eur J Radiol. 2016;85(10):1716-21.
  9. Baumgartner RN, Koehler KM, Gallagher D, Romero L, Heymsfield SB, Ross RR, et al. Epidemiology of sarcopenia among the elderly in New Mexico. Am J Epidemiol. 1998;147(8):755-63.
  10. Schober P, Boer C, Schwarte LA. Correlation Coefficients: Appropriate Use and Interpretation. Anesthesia & Analgesia. 2018;126(5):1763-8.

Reviewer 2 Report

Thank you for your revision.

I believe the authors replied to my comments.

As for the question R2Q2, the target image quality is not SNR or CNR in the obtained images.  These are the values that are preset on the CT console before the examination. Usually, it is SD for CANON, CNR for Siemens, noise index for GE, etc. For example, target SD of abdominal CT in my institute is set at the value of 15 for CANON CT by using the AEC algorithm.

Author Response

Dear reviewer,

Thank you for providing second feedback on our manuscript. We appreciate the comments and believe that they have helped us improve the manuscript.
